# Safety of Co-Administration Versus Separate Administration of the Same Vaccines in Children: A Systematic Literature Review

**DOI:** 10.3390/vaccines8010012

**Published:** 2019-12-31

**Authors:** Jorgen Bauwens, Luis-Henri Saenz, Annina Reusser, Nino Künzli, Jan Bonhoeffer

**Affiliations:** 1Faculty of Medicine, University of Basel, 4056 Basel, Switzerland; 2University of Basel Children’s Hospital, 4056 Basel, Switzerland; 3Brighton Collaboration Foundation, 4056 Basel, Switzerland; 4Swiss Tropical and Public Health Institute, 4056 Basel, Switzerland

**Keywords:** children, minors, vaccination, vaccines, safety, adverse effects, co-administration

## Abstract

The growing number of available vaccines that can be potentially co-administered makes the assessment of the safety of vaccine co-administration increasingly relevant but complex. We aimed to synthesize the available scientific evidence on the safety of vaccine co-administrations in children by performing a systematic literature review of studies assessing the safety of vaccine co-administrations in children between 1999 and 2019, in line with Preferred Reporting Items for Systematic Reviews and Meta-Analyses (PRISMA) guidelines. Fifty studies compared co-administered vaccines versus the same vaccines administered separately. The most frequently studied vaccines included quadrivalent meningococcal conjugate (MenACWY) vaccine, diphtheria and tetanus toxoids and acellular pertussis (DTaP) or tetanus toxoid, reduced diphtheria toxoid and acellular pertussis (Tdap) vaccines, diphtheria and tetanus toxoids and acellular pertussis adsorbed, hepatitis B, inactivated poliovirus and *Haemophilus influenzae* type b conjugate (DTaP-HepB-IPV/Hib) vaccine, measles, mumps, and rubella (MMR) vaccine, and pneumococcal conjugate 7-valent (PCV7) or 13-valent (PCV13) vaccines. Of this, 16% (n = 8) of the studies reported significantly more adverse events following immunization (AEFI) while in 10% (n = 5) significantly fewer adverse events were found in the co-administration groups. Statistically significant differences between co-administration and separate administration were found for 16 adverse events, for 11 different vaccine co-administrations. In general, studies briefly described safety and one-third of studies lacked any statistical assessment of AEFI. Overall, the evidence on the safety of vaccine co-administrations compared to separate vaccine administrations is inconclusive and there is a paucity of large post-licensure studies addressing this issue.

## 1. Introduction

With new vaccines becoming available and added to pediatric immunization schedules, these schedules become increasingly crowded [1,2]. Since co-administering vaccines may facilitate the introduction of new vaccines to immunization schedules and positively affect coverage rates [3], a growing number of vaccines is likely to be co-administered in the future. Uncertainty about the safety of co-administered vaccines can contribute to vaccine hesitancy in parents [4,5]. This highlights the need for assessing the safety of co-administered vaccines.

Immunization schedules are typically designed based on evidence of efficacy and safety from clinical trials. However, both the number and types of vaccines co-administered in routine immunization practices, as well as the vaccinated populations, may differ from the ones investigated in pre-licensure trials. In addition, the small sample size of clinical trials, the many possibilities of vaccine co-administrations, and the low incidence of adverse events following immunization (AEFI) make it challenging to find and interpret evidence on the safety of vaccine co-administrations compared to separate administrations. Both healthcare providers and parents require more information about vaccine co-administrations [6]. Therefore, we performed a systematic literature review, aiming to synthesize the available scientific evidence on the safety of vaccine co-administrations in children.

## 2. Methods

We performed a systematic literature review of studies assessing the safety of vaccine co-administrations in children in line with Preferred Reporting Items for Systematic Reviews and Meta-Analyses (PRISMA) guidelines. Our search strategy aimed to retrieve studies in the pediatric population, who received more than one vaccine at the same time for which adverse outcomes were reported. We searched Pubmed (including Medline), Embase, and the Cochrane library for articles in English, published between 1999 and 2019 to cover vaccines and co-administrations relevant to actual immunization practices, combining the following keywords:Population: Infant OR child OR adolescent OR newborn OR minors OR teenager;Intervention: Vaccination OR vaccines OR immunization OR immunization schedule OR immunization OR immunization, secondary/trends OR mass vaccination/trends OR vaccines/administration & dosage OR vaccines/pharmacology; andOutcome: ((Safety drug-related side effects OR adverse reactions OR adverse effects OR vaccination/adverse effects OR vaccines/complications OR vaccines/adverse effects) OR safety OR tolerability) AND (co-administration OR co-administered OR concomitant administration OR simultaneous administration).

This translated in the following search string for Pubmed: “(Infant OR child OR adolescent OR newborn OR minors [MeSH terms]) AND (vaccination OR vaccines OR immunization OR immunization schedule OR immunization OR immunization, secondary/trends OR mass vaccination/trends OR vaccines/administration and dosage OR vaccines/pharmacology [MeSH terms]) AND ((safety drug-related side effects OR adverse reactions OR adverse effects OR vaccination/adverse effects OR vaccines/complications OR vaccines/adverse effects [MeSH terms]) OR safety) AND (co-administration OR co-administered OR concomitant administration OR simultaneous administration)”. The most recent search was performed on 28 January 2019. We screened the included articles’ reference lists for additional articles. Full text articles were obtained through the University of Basel’s library and references were managed using Zotero [7].

Articles were eligible when study participants were under 18 years of age or the study population included both under and over 18 year olds, co-administration of at least two vaccines was indicated in the title and/or abstract, and safety data were reported. After removing duplicates, the following data was collected from the included articles by three independent reviewers (J.B. (Jorgen Bauwens), L.-H.S., A.R.). Study population: Minimum and maximum age of children included, sample size, selected inclusion and exclusion criteria applied (i.e., subpopulations, conditions leading to exclusion). Intervention: Vaccines co-administered and comparator vaccines. Outcome: All AEFIs observed, reported differences in AEFI between co-administration and comparator groups (i.e., statistically significantly more or fewer AEFI, more or fewer AEFI without statistical assessment). Study characteristics: Study design, countries, statistics reported to assess differences in AEFI, potential sources of bias. Coding, completeness, and consistency of variables in the data extraction forms were checked among the reviewers and data were compiled in a structured database.

The safety assessment was limited to studies comparing co-administered vaccines with the same vaccines administered separately. Studies where the comparator group did not receive exactly the same vaccines separately as the vaccines co-administered were excluded. For studies comparing co-administration with separate administration of the same vaccines, the collected data was analyzed to obtain the following summary measures: Vaccines investigated in co-administration versus separate administration of the same vaccines and their frequencies of occurrence among the included studies; frequencies of study designs used to assess co-administration versus separate administration; minimum, maximum, and mean sample sizes of the included studies by study design; minimum and maximum ages of children in the included studies; number of studies per country; number of studies with statistically more or statistically fewer AEFI between both groups, number of studies with more or fewer AEFI between both groups without statistical assessment provided; AEFI that were reported statistically significantly more or less between both study groups; use of statistical measures in the included studies; occurrence of potential sources of bias including health status of the study population, exclusion of children with known previous reaction or allergies to vaccines or vaccine components; and method for reporting and collecting AEFIs. Analyses were performed in R [8].

## 3. Results

From 391 retrieved articles, 185 studies reported safety data for co-administration of at least two vaccines in children. Of these, 50 studies (27%) compared co-administration with separate administration of the same vaccines and were included in our analysis. Other studies meeting the initial inclusion criteria, but not allowing a direct comparison between co-administration and separate administration, compared co-administered vaccines versus only a part of the same vaccines administered separately (n = 56, 30%), versus the same antigens but combined in one vaccine (n = 20, 11%), versus other vaccines (n = 6, 3%), or looked at co-administered vaccines without comparison (n = 58, 31%). Figure 1 displays the study attrition diagram.

### 3.1. Vaccines Studied

The most frequently investigated co-administered vaccines included MenACWY vaccine (n = 16, 32%), DTaP or Tdap vaccines (n = 11, 22%), DTaP-HepB-IPV/Hib vaccine (n = 10, 20%), MMR vaccine (n = 9, 18%), and PCV7 or PCV13 vaccines (n = 9, 18%) (Figure 2). Appendix A provides an overview of the study characteristics and findings of all studies comparing co-administered vaccines versus the same vaccines administered separately. The full meaning of vaccine abbreviations can be found in Appendix A.

### 3.2. Study Characteristics

The median sample size of the 50 studies comparing co-administration with separate administration of the same vaccines was 726 (interquartile range (IQR) 328-1199). Forty-five (90%) of these studies were randomized clinical trials with a sample size between 64 and 2648 children. One case-control study included 590 children. One prospective observational study had a study size of 530 children and one retrospective observational study included 36,844 children. One study used surveillance data covering 128,197 vaccinations and one study used case reports from 883 children. Table 1 lists the sample sizes of these trials by phase. The minimum ages of children enrolled in the studies varied between birth and 16 years (median 1 year) and the maximum ages of the enrolled study population varied between 7 weeks and 49 years (median 23 months). Seven studies (14%) also included persons over 18 years whose data were deemed relevant for assessing the safety of co-administration and were therefore included in our analysis. Figure 3 shows the geographic distribution of these studies and highlights that studies were particularly conducted in the US and Europe.

Only healthy children were enrolled in 37 (74%) studies and 20 studies (40%) excluded children with known allergies or hypersensitivity to vaccines or vaccine components. In 37 (74%) studies, the safety data relied on parental self-reporting of AEFI.

### 3.3. Safety Outcomes

Thirteen (26%) studies comparing co-administered vaccines with the same vaccines administered separately found statistically significant safety differences. Of these, eight studies (16%) reported significantly more and five studies (10%) reported significantly fewer AEFI in the respective co-administration groups. Of the eight studies identifying significantly more AEFI, two found significant increases in pyrexia: One when co-administering PCV13 + IIV3 (RD: 20.6%, RR: 2.2) and one when co-administering DTaP-HepB-IPV/Hib + PCV7 (RD: 14.7%, RR: 2.5) compared to separate administration of these vaccines [9,10]. One study reported significant increases in injection site pain (risk difference (RD): 6.3%, relative risk (RR): 1.1) and injection site bruising (RD: 3.6%, RR: 2.6) when co-administering MenACWY + Tdap + HPV compared to separate administration [11]. One study reported significant increases in injection site swelling (RD: 5.0%, RR: 1.5) when co-administering MenACWY + Tdap + HPV compared to separate administration [12], and one study reported a significant increase in myalgia (RD: 16%, RR: 1.5) after co-administering MenACWY + Tdap + HPV [13]. One study reported significant increases of injection site tenderness (RD: 15.6%, RR: 2.7) and headache (RD: 22.9%, RR: 3.7) after co-administering Td + MMR + HepB compared to separate administration [14]. One study reported a significant increase in vomiting (RD: 10.0%, RR: 2.0) following DTaP-IPV/Hib + MenC + RV5 co-administration [15], and one study reported significant increases in overall adverse events following co-administration (RD: 19.1%, RR: 1.5) of DTaP-IPV/Hib + MMR compared to their separate administrations [16].

Of the five studies identifying significantly fewer AEFI, one study reported significantly less diarrhea (RD: −20.3%, RR: 0.5) and pyrexia (RD: −11.3%, RR: 0.5) following co-administration of DTaP-IPV + RV5 [17]. One study reported significantly less injection site erythema (RD: −15.4%, RR: 0.7) following DTaP-HepB-IPV/Hib + MenC co-administration [18]. One study reported significantly less rash (RD: −5.8%, RR: 0.6) and rhinorrhea (RD: −6.1%, RR: 0.7) after + MMR + VAR + Hib-HepB co-administration compared to separate administration [19]. One study reported significantly less nasopharyngitis (RD: −3.5%, RR: 0.6) and insomnia following co-administering PCV7 + MMRV compared to separate administration [20]. One study reported significantly less conjunctivitis (RD: −0.7%, RR: 0.1) after co-administering OPV and LAIV compared to separate administration [21]. The reported incidences of AEFIs are presented in Figure 4. Appendix A provides a summary of the major study characteristics.

Thirty-three (66%) of studies comparing co-administered vaccines versus the same vaccines administered separately reported safety differences without providing a statistical assessment: In 29 (58%) of these studies increased AEFI were found in the co-administration groups and in 17 (34%) of these studies decreased AEFI were found in the co-administration groups.

Risk of AEFI and differences between groups were statistically evaluated and reported in studies comparing co-administration with separate administration of the same vaccines by assessing confidence intervals (48%), p-values (28%), risk differences (10%), relative risks (4%), Fisher test (2%), adjusted relative risk (aRR) (2%), IR (1%), or odds ratios (2%). Seventeen (34%) studies reported no statistical assessment. Of those, two studies (4%) listed observed AEFI without reporting absolute numbers or percentages.

## 4. Discussion

The evidence about the safety of co-administered vaccines compared to separately administered vaccines is mainly based on clinical trials that were primarily designed to evaluate efficacy rather than safety differences. The safety of co-administering vaccines was assessed in 185 studies over the last 20 years. Of these, only 50 directly compared the safety of co-administration with separate administration of the same vaccines. Most occurred in Europe and the USA, reflecting the regions where the most clinical trials take place [22] and where databases with observational data are available. The remaining 135 studies assessed the safety of co-administration and revealed safety data but did not allow a comparison with separate administration because they lacked a control group who received the same antigens as separate vaccines. The control groups in these studies received fewer antigens, received the antigens in a combined vaccine, received other antigens, or the control group did not receive any vaccine.

For the majority of co-administered vaccines, only one study directly assessing the safety of vaccine co-administration versus separate administration was available. Co-administrations of MenACWY + Tdap [11,12,13,23], and MenACWY + Tdap + HPV [24,25,26,27] were studied in four different trials each. Co-administrations of DTaP-HepB-IPV/Hib + PCV [9,28,29], DTaP-HepB-IPV/Hib + MMRV [30,31,32], and MMR + VAR [33,34,35] were studied in three different studies each. Co-administrations of MenACWY + DTaP-HepB-IPV/Hib [36,37], DTaP-IPV/Hib + MMR [16,32], HPV + HepB [38,39], and IIV (H1N1) + IIV3 [40,41] were each evaluated in two studies.

We only found statistically significant differences between co-administration and separate administration for some AEFI, and for a limited number of vaccine co-administrations. Furthermore, multiple studies on the same co-administered vaccines did not confirm each other’s findings, as indicated in Table 2. Despite much more injection site bruising and slightly more injection site pain after co-administering MenACWY + Tdap + HPV found in one study [11], three other studies evaluating the same co-administered vaccines [12,13,23] could not confirm this increase. On the other hand, one of these studies detected an increase in myalgia after co-administering MenACWY + Tdap + HPV [13] but the three similar studies did not [11,12,23]. Likewise, only one of these studies found an increase of injection site swelling after co-administration [12] in contrast to the others [11,13,23]. Nevertheless, the incidence rates of these adverse events were in line with those reported in a study investigating the co-administration of MenACWY + Tdap + HPV but without a separate administration control group [42]. Similarly, only one of three studies on DTaP-HepB-IPV/Hib + PCV7 found a strong increase in cases of pyrexia after co-administration [9,28,29]. Also, here the incidence rates of fever were comparable with those observed in six other studies investigating the co-administration of the same vaccines but without a separate administration control group [43,44,45,46,47,48]. The consistency in incidence rates indicates that the observations are reliable and that the failure to detect significant differences rather might be due to a lack of statistical power.

For the co-administered vaccines where only one study could be retrieved, almost three times more cases of injection site tenderness and almost four times more headaches were reported following co-administration of Td + MMR + HepB [14], more than twice as many cases of pyrexia were found after co-administering PCV13 + IIV3 [10], and twice as much vomiting was reported following co-administering DTaP-IPV/Hib + MenC + RV [15]. A smaller increase in overall adverse events following co-administration of DTaP-IPV/Hib + MMR was observed in the only study with these vaccines [16].

Some studies found fewer AEFI after co-administration compared to separate administration: Half the cases of diarrhea and half the cases of pyrexia following co-administration of DTaP-IPV + RV5 [17], less injection site erythema following DTaP-HepB-IPV/Hib + MenC co-administration [18], almost half as much rash and less rhinorrhea after MMR + VAR + Hib-HepB co-administration [19], almost half as much nasopharyngitis and less insomnia following co-administering PCV7 + MMRV co-administration [20], and less conjunctivitis after co-administering OPV + LAIV compared to separate administration [21]. All these co-administrations were assessed in only one study each.

Despite the few studies on the same co-administered vaccines, it is remarkable that none of the reported increased adverse events following co-administration were contradicted by another study that would report a significant decrease following the same co-administration, and vice versa. In general, the studies indicate differences in less severe AEFI. Therefore, these insights might not influence immunization practices that much (also because we have not addressed the potential impact of co-administration on efficacy in our review) but can be meaningful to correctly inform patients and parents.

The lack of repeated studies for the majority of vaccine co-administrations and the absence of confirmatory findings of significant results indicate a scarcity of strong evidence about the safety of co-administration versus separate administration. This lack of evidence can be partly explained by the inability to demonstrate statistically significant safety differences. Two-thirds of studies reported differences in safety between vaccine co-administration and separate administration but these were not significant or a statistical assessment was missing. Typically, safety was briefly described and one-third of studies lacked any statistical assessment of AEFI. Most of the studies were randomized clinical trials (RCTs) mainly designed to demonstrate efficacy, with sample sizes that were too small for assessing particularly rare and very rare adverse events with sufficient statistical power [49]. This may be a reason why studies failed to detect statistically significant differences in safety. Observational studies with larger sample sizes assessing co-administration versus separate administration have better potential to achieve sufficient statistical power. However, such studies were found to be rare. Publication bias towards publishing studies with a positive benefit–risk balance may also affect the availability of information on safety and hence affect the findings of our review. Studies with an unsatisfactory immunogenicity and/or an unfavorable safety profile might not have been published.

Our findings indicate that dedicated studies on vaccine co-administration with a larger sample size are required to obtain statistical evidence on a potential increase or decrease of adverse events following co-administration. Particularly for co-administered vaccines for which an increased or decreased risk compared to separate administration was observed, confirmatory studies specifically designed to assess the safety of co-administration would be useful. Such studies should aim at assessing AEFIs with sufficient statistical power and would benefit from standardized data collection of AEFIs and established methodologies for the assessment of adverse events following vaccine co-administration compared to separate administration.

## 5. Conclusions

Evidence about no increased risk of adverse events when co-administering vaccines compared to separate vaccine administration is indispensable to improve immunization rates. Opposition to vaccination and under-vaccination are crucial threats to herd immunity [50], which can be addressed by proving the safety of vaccine co-administration. Co-administration is an efficient vaccination strategy, associated with high coverage rates [3] and vaccine timeliness [51]. While there is no indication to be concerned about the safety of co-administered vaccines, healthcare providers must aim for the highest standards of care. Particularly for preventive care in children such as immunization, we must aim for the best strategies that entail the lowest risks. Considering the scale of immunizing children and vaccine co-administrations in real life, the currently available evidence is limited and inconclusive. This study indicated that differences in safety of vaccine co-administrations compared to separate vaccine administrations may exist, particularly for more common, less severe AEFI. However, based on the currently available evidence, it is challenging to verify the true extent and impact. In summary, there is limited and inconclusive evidence available about the difference in safety of vaccine co-administrations compared to separate vaccine administrations in children.

## Figures and Tables

**Figure 1 vaccines-08-00012-f001:**
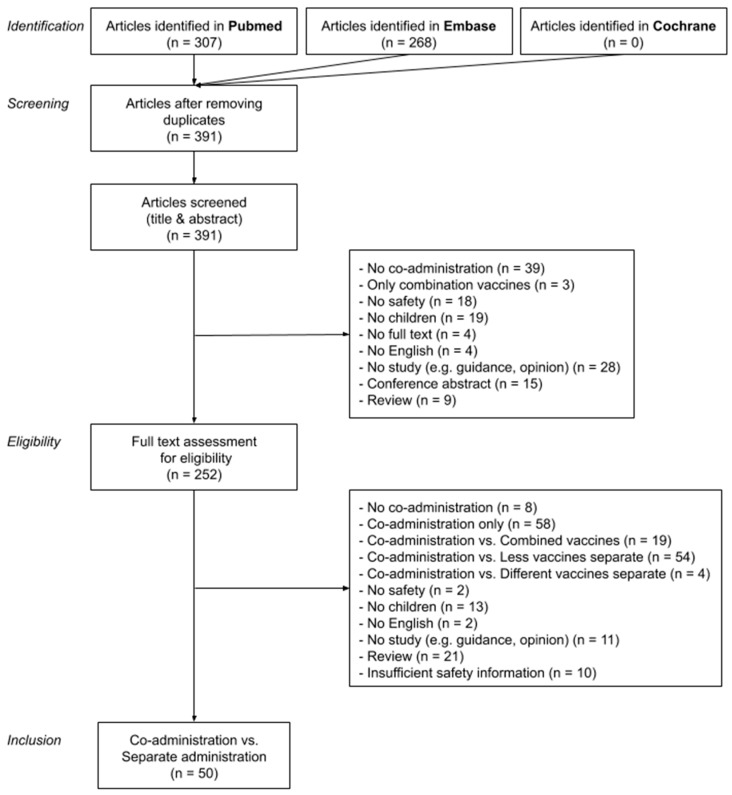
Flow diagram of identifying, screening, assessing eligibility, and including studies.

**Figure 2 vaccines-08-00012-f002:**
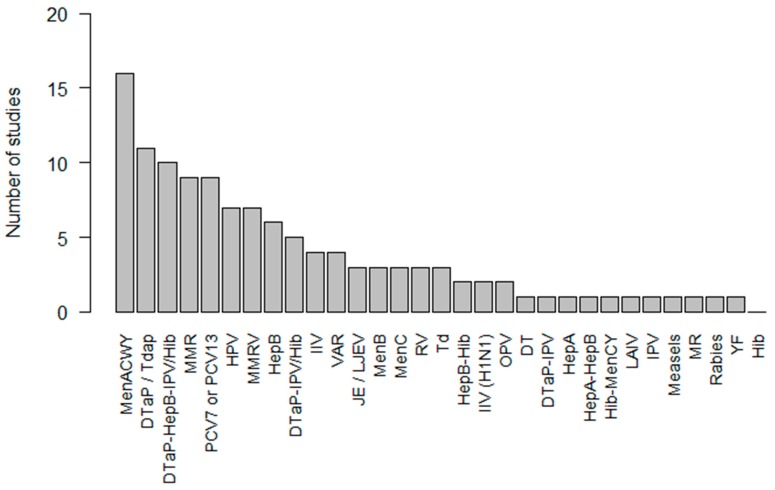
Frequency of vaccines investigated in co-administration versus separate administration studies.

**Figure 3 vaccines-08-00012-f003:**
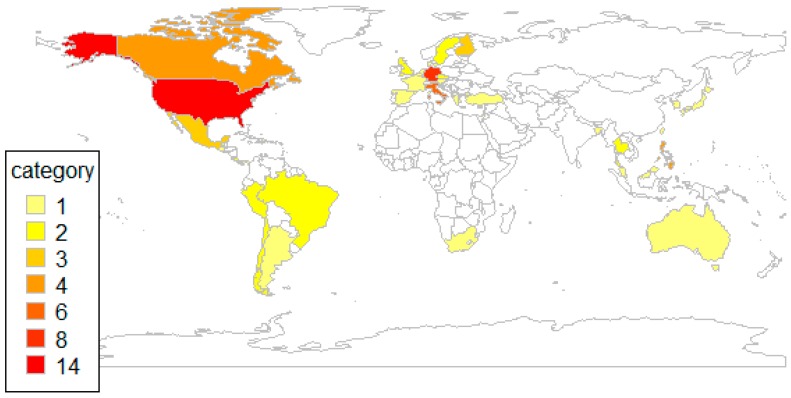
Geographical distribution of studies comparing co-administration versus separate administration.

**Figure 4 vaccines-08-00012-f004:**
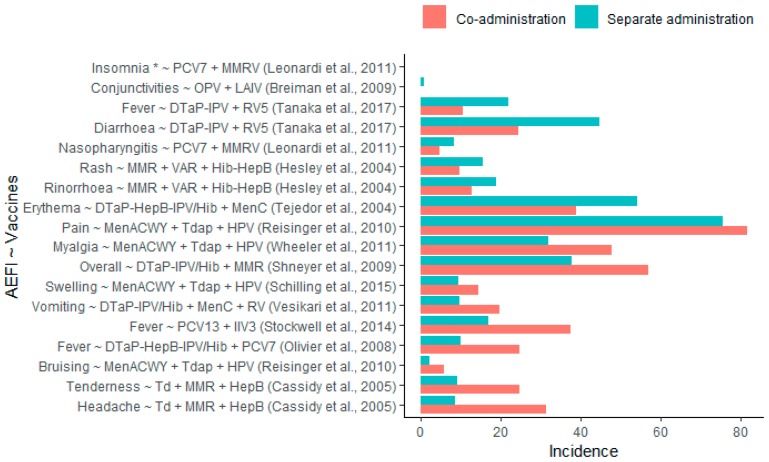
Incidences of adverse events following immunization (AEFIs) with statistically significant differences following co-administration compared to separate administration. *No incidences reported.

**Table 1 vaccines-08-00012-t001:** Sample sizes by study type.

**Study Type**	**n**	**Minimum**	**Sample Size Median**	**Maximum**
RCT (no phase specified)	27	64	550	2503
RCT phase 2	3	200	2499	2648
RCT phase 2b	1		460	
RCT phase 3	9	312	802	1620
RCT phase 3b	2	716	730	744
RCT phase 4	3	376	1341	1504
Case Control	1		590	
Prospective Observational Cohort	1		530	
Retrospective Observational Cohort	1		36,844	
Surveillance report	1		128,297	
Case Reports	1		833	

**Table 2 vaccines-08-00012-t002:** Number of studies with statistically significant differences in AEFI after co-administration compared to separate administration.

Vaccines Co-Administered	Number of Studies	AEFI	Stat. Sign. More AEFI	Stat. Sign. Fewer AEFI	No Stat. Sign. Difference
DTaP-HepB-IPV/Hib + MenC	1	Injection site erythema	0 (0%)	1 (100%)	0 (0%)
DTaP-HepB-IPV/Hib + PCV7	3	Pyrexia	1 (33%)	0 (0%)	2 (67%)
DTaP-IPV + RV5	1	Diarrhoea	0 (0%)	1 (100%)	0 (0%)
		Pyrexia	0 (0%)	1 (100%)	0 (0%)
DTaP-IPV/Hib + MenC + RV	1	Vomiting	1 (100%)	0 (0%)	0 (0%)
DTaP-IPV/Hib + MMR	1	Overall	1 (100%)	0 (0%)	0 (0%)
MenACWY + Tdap + HPV	4	Injection site bruising	1 (25%)	0 (0%)	3 (75%)
		Injection site pain	1 (25%)	0 (0%)	3 (75%)
		Injection site swelling	1 (25%)	0 (0%)	3 (75%)
		Myalgia	1 (25%)	0 (0%)	3 (75%)
MMR + VAR + Hib-HepB	1	Rash	0 (0%)	1 (100%)	0 (0%)
		Rhinorrhoea	0 (0%)	1 (100%)	0 (0%)
OPV + LAIV	1	Conjunctivitis	0 (0%)	1 (100%)	0 (0%)
PCV7 + MMRV	1	Insomnia	0 (0%)	1 (100%)	0 (0%)
		Nasopharyngitis	0 (0%)	1 (100%)	0 (0%)
PCV13 + IIV3	1	Pyrexia	1 (100%)	0 (0%)	0 (0%)
Td + MMR + HepB	1	Headache	1 (100%)	0 (0%)	0 (0%)
		Injection site tenderness	1 (100%)	0 (0%)	0 (0%)

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
