# Peer review of "Safety of Co-Administration Versus Separate Administration of the Same Vaccines in Children: A Systematic Literature Review"

_vaccines, 2019, doi:10.3390/vaccines8010012_

Round 1
Reviewer 1 Report
This is a well researched and well written manuscript. I have a few minor suggestions;
Line 148, 203 - IIV3 is not mentioned in the list of abbreviations.
Line 131 states that 7 studies included persons over 18 years of age and was included in analysis, but this disputes what is mentioned on Line 73, where it states "articles were eligible when study participants were under 18 years of age". I would mention a clause stating this exception.
Lines 198-204 mentions about safety assessment and co-administration of different vaccines in multiple studies, but does not include any citation references.
Lines 206-212, the statements mentioned is unclear and is different than what is mentioned in Table 2. I would re-phrase the sentences, and would consider including certain graphical representation of the AEFI statistical significance, as this manuscript includes extensive statistical analyses researching various papers.
Author Response
Dear Reviewer
Thank you for reviewing our paper and for your positive feedback on our research!
We appreciate your comments which were helpful to further improve our paper. Please find our replies to your comments below:
1. Line 148, 203 - IIV3 is not mentioned in the list of abbreviations.
--> Added to the list
2. Line 131 states that 7 studies included persons over 18 years of age and was included in analysis, but this disputes what is mentioned on Line 73, where it states "articles were eligible when study participants were under 18 years of age". I would mention a clause stating this exception.
--> We clarified by adding the text in bold: "Articles were eligible when study participants were under 18 years of age or the study population included both under and over 18 year olds […]"
3. Lines 198-204 mentions about safety assessment and co-administration of different vaccines in multiple studies, but does not include any citation references.
--> References added
4. Lines 206-212, the statements mentioned is unclear and is different than what is mentioned in Table 2. I would re-phrase the sentences, and would consider including certain graphical representation of the AEFI statistical significance, as this manuscript includes extensive statistical analyses researching various papers.
--> The text lists the co-administrations that were studied in more than one trial. We have slightly modified the text.
Table 2 doesn’t list all the studies as described in this paragraph and is illustrative for the next paragraph of the discussion. This table lists the co-administrations (including the number of studies) for which significant differences in AEFI were reported. We opted for a table here because we could provide some more details this way. The underlying data was visualised for the results in Figure 4.
We hope that we have addressed your comments in a way that is acceptable for you.
Kind regards
Jorgen Bauwens
Reviewer 2 Report
This is well-written and thorough review article that highlights the importance of identifying the risks and benefits of vaccine co-administration versus administration of fewer matched vaccines per visit. The tables presented summarize complex text found in the text, which makes the review process very straight-forward. This article will be suitable for publication after the authors address some minor points. Such changes will further strengthen the article.
The authors should provide a list of abbreviations that would include (but is not limited to) all vaccines discussed in the article.
In the first line of the Discussion, the authors mention that many co-administration studies focus on vaccine efficacy and not so many on safety. Are there any review articles in the literature base that discuss efficacy of vaccine co-administration that the authors can reference? Adding just a little bit of text and a handful of references would be of particular interest to researchers who focus on the immune responses behind vaccination. This could help to broaden the audience of readers for this interesting review article.
It would be great if the authors could add their collective thoughts on any patterns that they observe regarding the combination of vaccines administered and increased/decreased adverse events. Such information is presented throughout the review article text, but including a take-home message of the authors' collective thoughts on any patterns would help to round out this nice review article.
Perhaps not surprisingly, most of the studies that were included in the final analysis were from the US and Europe. While perhaps beyond the scope of this review article, it would be interesting if the authors could mention why Asia and Africa do not have many studies that focus on vaccine safety studies. Things that come to the reviewer's mind are lack of national vaccination programs or lack of adverse events databases from countries in Asia and Africa. There are certainly others that could be mentioned.
Are there any articles in the literature (either the 50 studies included in this review article or others in the literature) that discuss which clinical variables could increase or decrease the likelihood of an increased or decreased risk of AEFI (e.g. sex, race, etc)?
The reviewer found just a handful of typos:
Line 207 "Table 2" should be capitalized.
Line 16, there is an extra space "and a half times"
Line 171 "Figure 4" should be capitalized.
Author Response
Dear Reviewer
Thank you for reviewing our paper and for your positive feedback on our work!
We greatly appreciate your thoughts and comments which were interesting to read and helpful to further improve our paper. Please find our replies to your comments below:
1. The authors should provide a list of abbreviations that would include (but is not limited to) all vaccines discussed in the article.
--> Some missing abbreviations have been added to the list that can be found as a supplementary table (S2).
2. In the first line of the Discussion, the authors mention that many co-administration studies focus on vaccine efficacy and not so many on safety. Are there any review articles in the literature base that discuss efficacy of vaccine co-administration that the authors can reference? Adding just a little bit of text and a handful of references would be of particular interest to researchers who focus on the immune responses behind vaccination. This could help to broaden the audience of readers for this interesting review article.
--> A complete evaluation of immunisation should indeed consider the balance between efficacy and safety, but this was not the purpose of our study. There are indeed review articles on the efficacy (and safety) of specific co-administered vaccines. However, we are concerned that, given the rather broad scope of included vaccines and AEFI, adding meaningful information on efficacy would add too much complexity to our review. In fact, all included and cited studies thoroughly addressed efficacy of co-administration. Thus, the best sources to learn about the corresponding efficacy of the discussed co-administered vaccines would be these studies.
3. It would be great if the authors could add their collective thoughts on any patterns that they observe regarding the combination of vaccines administered and increased/decreased adverse events. Such information is presented throughout the review article text, but including a take-home message of the authors' collective thoughts on any patterns would help to round out this nice review article.
--> We added a bit more meaning to our concluding paragraph by adding: “This study indicated that differences in safety of vaccine co-administrations compared to separate vaccine administrations may exist, particularly for more common, less severe events.” Our major concern is that the available studies were too small to detect significant differences, particularly for less common, major adverse events. We indeed discuss observed patterns in both directions (increases and decreases in more common, minor adverse events) throughout the text but these are not unambiguously. Therefore we feel not confident to make explicit statements on any patterns given the currently available information.
4. Perhaps not surprisingly, most of the studies that were included in the final analysis were from the US and Europe. While perhaps beyond the scope of this review article, it would be interesting if the authors could mention why Asia and Africa do not have many studies that focus on vaccine safety studies. Things that come to the reviewer's mind are lack of national vaccination programs or lack of adverse events databases from countries in Asia and Africa. There are certainly others that could be mentioned.
--> Good point and hence we have added the following in the discussion: “Most occurred in Europe and the USA, reflecting the regions where the most clinical trials take place [22] and where databases with observational data are available.”
5. Are there any articles in the literature (either the 50 studies included in this review article or others in the literature) that discuss which clinical variables could increase or decrease the likelihood of an increased or decreased risk of AEFI (e.g. sex, race, etc)?
--> Within the 50 included articles, this was not discussed specifically. However, studies exist that explore the influence of other factors, such as genetics. Since we focused on the difference between co-administration and separate administration in otherwise comparable groups (e.g. same population, age, mix of sex), these other variables should not bias the findings. Particularly since most studies were randomised trials. Although very interesting and therefore worth a review on itself, the influence of clinical factors was beyond the scope of our research.
6. The reviewer found just a handful of typos:
a. Line 207 "Table 2" should be capitalized.
--> corrected
b. Line 16, there is an extra space "and a half times"
--> corrected
c. Line 171 "Figure 4" should be capitalized.
--> corrected
We hope that we have addressed your comments in a way that is acceptable for you.
Kind regards
Jorgen Bauwens
Reviewer 3 Report
This study reviews published data pertaining to the safety of vaccine co-administration versus separate administration of the same vaccines in children/adolescents. It is an important contribution to an area where data are indeed lacking. The authors find overall that there is a paucity of reliable data, and published results indicate that there may be some differences in the safety profile of co-administration vs separate administration, but the numbers of events are insufficient to draw definitive conclusions.
Overall, this study is helpful in reviewing the available evidence on the safety of vaccine co-administration. I agree with the authors' assessment that it is important to prove that it is safe in order to combat vaccine hesitancy, especially since co-administration is an effective means to increase coverage rates. The same issue also applies in adults.
I have just a few comments for the authors' consideration (no particular order of importance):
1) I feel that the exclusion of 19 papers because you couldn't find the full text is quite a lot, compared to the final number of 50 studies included. There are ways of obtaining the full text, not least by contacting the authors directly. I really think that an additional effort could be made on this point, because the analysis of 69 papers (if all these unavailable texts were included) might yield different results.
2) In the same vein, the exclusion of a total of 6 papers because they were not in English is also problematic - while it is more comprehensible (especially if they were in a language that none of the authors can speak), it might also be worth checking again whether someone in your entourage could interpret the main results for you. Technology can help to shrink borders and remove language barriers.
3) Figure 3 - it is hard to distinguish between the shades of grey. Colour coding would be more legible, but this may not be possible for the authors - this is just a minor comment, at the author's discretion.
4) Lines 205-231 - To me, this reads very much like a results paragraph. What is the authors' interpretation? Taken together, what do these data mean for the international medical community? What are the implications? I think a more discussive text would be helpful.
5) Lines 248-249 - The authors state that papers with unsatisfactory immunogenicity may not have been published - this is hardly surprising. Mild fever or headache after vaccination may be tolerated in view of the health benefit yielded from avoiding serious disease, but a vaccine is certainly not likely to be tested large-scale if it is not only inefficacious for prevention of the disease, but also causes side effects!
6) Lines 265-267 - I would turn the authors' argument around to see it the other way around. Precisely because of the large-scale ongoing immunisation of children around the world, I would posit that the lack of data is in itself an indication of the frequency of safety concerns. If major adverse events were happening frequently, they would surely come to light sooner with the number of children being immunised around the world. However, I agree that this is certainly not an argument to say that there are no safety concerns.
7) I think the authors should mention in the limitations that they did not look at the efficacy of the co-administration versus separate administration for the prevention of disease.
8) Finally, I ran your search terms through PubMed and found 349 results, only 2 of which were published since January when the authors ran their search. The authors indicate that they found 283 results in PubMed - there may be some confusion regarding the exact terms of the search? The authors might like to verify this point.
9) Overall, the paper is well written, but there are some grammatical errors that warrant correction (e.g. several instances of "less events", should be "fewer events", to cite but one example).
Author Response
Dear Reviewer
Thank you for reviewing our paper and for your positive feedback on our research!
We greatly appreciate your comments which we found interesting and helpful to further improve our paper. Please find our replies to your comments below:
1) I feel that the exclusion of 19 papers because you couldn't find the full text is quite a lot, compared to the final number of 50 studies included. There are ways of obtaining the full text, not least by contacting the authors directly. I really think that an additional effort could be made on this point, because the analysis of 69 papers (if all these unavailable texts were included) might yield different results.
--> Fifteen of these 19 references concerned conference abstracts and we have now adapted the flowchart with this more specific information. We indeed tried to retrieve the four remaining publications through personal contacts, but were not successful. These abstracts did not include sufficient information to include in our review. However, we noticed that several of these abstracts (typically reporting intermediate results) were followed by full publications later. These papers were then included when matching our inclusion criteria.
2) In the same vein, the exclusion of a total of 6 papers because they were not in English is also problematic - while it is more comprehensible (especially if they were in a language that none of the authors can speak), it might also be worth checking again whether someone in your entourage could interpret the main results for you. Technology can help to shrink borders and remove language barriers.
--> Four of these six papers would also have been excluded due to not meeting our inclusion criteria (three not being studies, one lacking separate administration data). Bearing in mind that our targeted readers (Vaccines publishing in English), we believed that language might be a significant barrier and therefore listed these papers under “no English”. It could also be appropriate to list these under the other applicable exclusion criteria and we can do so if you prefer. For now, we left it as initially reported since we had chosen English as an inclusion criterion. For the same reason, we would not include the two remaining articles. An additional reason for this is that the (small, local) publishing journals might not apply the same quality criteria as we expect from international peer-review journals.
3) Figure 3 - it is hard to distinguish between the shades of grey. Colour coding would be more legible, but this may not be possible for the authors - this is just a minor comment, at the author's discretion.
--> Figure replaced by colour figure.
4) Lines 205-231 - To me, this reads very much like a results paragraph. What is the authors' interpretation? Taken together, what do these data mean for the international medical community? What are the implications? I think a more discussive text would be helpful.
--> We have re-written this part more as a discussion and we have elaborated including some more interpretation and reflection on implications.
5) Lines 248-249 - The authors state that papers with unsatisfactory immunogenicity may not have been published - this is hardly surprising. Mild fever or headache after vaccination may be tolerated in view of the health benefit yielded from avoiding serious disease, but a vaccine is certainly not likely to be tested large-scale if it is not only inefficacious for prevention of the disease, but also causes side effects!
--> This is absolutely true. We didn’t elaborate on this because our focus was the difference in safety between vaccine co-administration and separate administration, without considering efficacy. A complete evaluation of immunisation should indeed consider the balance between efficacy and safety, but this was not the purpose of our study.
6) Lines 265-267 - I would turn the authors' argument around to see it the other way around. Precisely because of the large-scale ongoing immunisation of children around the world, I would posit that the lack of data is in itself an indication of the frequency of safety concerns. If major adverse events were happening frequently, they would surely come to light sooner with the number of children being immunised around the world. However, I agree that this is certainly not an argument to say that there are no safety concerns.
--> We agree that the turned argument likely makes sense. However, this “indication of safety by the absence of proof of major adverse events” might be insufficient to tackle vaccine hesitancy. Since we want to address exactly this issue, we prefer to keep the argumentation in its original way.
7) I think the authors should mention in the limitations that they did not look at the efficacy of the co-administration versus separate administration for the prevention of disease.
--> We have included this: “Therefore, these insights might not influence immunisation practices that much – also because we have not addressed the potential impact of co-administration on efficacy in our review – but can be meaningful to correctly inform patients and parents.”
8) Finally, I ran your search terms through PubMed and found 349 results, only 2 of which were published since January when the authors ran their search. The authors indicate that they found 283 results in PubMed - there may be some confusion regarding the exact terms of the search? The authors might like to verify this point.
--> Thank you very much for checking! The reported terms were correct but the listed number was false. Please note that we found 307 records (corrected in figure 1) instead of 349, which seems due to our applied limits (> 1 January 1999; Humans, English).
9) Overall, the paper is well written, but there are some grammatical errors that warrant correction (e.g. several instances of "less events", should be "fewer events", to cite but one example).
--> Corrected the indicated mistakes and reviewed the text for other grammatical errors.
We hope that we have addressed your comments in a way that is acceptable for you.
Kind regards
Jorgen Bauwens